# CAiD: Context-Aware Instance Discrimination for Self-supervised Learning in Medical Imaging

**Mohammad Reza Hosseinzadeh Taher**[1]               mhossei2@asu.edu

**Fatemeh Haghighi**[1]                                fhaghigh@asu.edu

**Michael B. Gotway**[2]                        Gotway.Michael@mayo.edu

**Jianming Liang**[1]                           jianming.liang@asu.edu

[1] *Arizona State University, AZ, USA*
[2] *Mayo Clinic, AZ, USA*

**Editors:** Under Review for MIDL 2022

## Abstract

Recently, self-supervised instance discrimination methods have achieved significant success in learning visual representations from unlabeled photographic images. However, given the *marked* differences between photographic and medical images, the efficacy of instance-based objectives, focusing on learning the most discriminative global features in the image (i.e., wheels in bicycle), remains unknown in medical imaging. Our preliminary analysis showed that high global similarity of medical images in terms of anatomy hampers instance discrimination methods for capturing a set of distinct features, negatively impacting their performance on medical downstream tasks. To alleviate this limitation, we have developed a simple yet effective self-supervised framework, called Context-Aware instance Discrimination (**CAiD**). CAiD aims to improve instance discrimination learning by providing finer and more discriminative information encoded from a diverse local context of unlabeled medical images. We conduct a systematic analysis to investigate the utility of the learned features from a three-pronged perspective: (i) generalizability and transferability, (ii) separability in the embedding space, and (iii) reusability. Our extensive experiments demonstrate that CAiD (1) enriches representations learned from existing instance discrimination methods; (2) delivers more discriminative features by adequately capturing finer contextual information from individual medial images; and (3) improves reusability of low/mid-level features compared to standard instance discriminative methods. As open science, all codes and pre-trained models are available on our GitHub page: https://github.com/JLiangLab/CAiD.
**Keywords:** Self-supervised Learning, Instance Discrimination, Transfer Learning.

## 1. Introduction

Self-supervised learning (SSL) aims to learn general-purpose representations without relying on human-annotated labels. Self-supervised *instance discrimination* methods (Chen et al., 2020a; Grill et al., 2020; He et al., 2020), which treat each image as a separate class, have rapidly closed the performance gap with supervised pre-training in various vision tasks (Gansbeke et al., 2021; Chen et al., 2021c). However, most existing instance discrimination methods are still primarily trained and evaluated on photographic images; therefore, their effectiveness and limitations for medical imaging applications remain unclear.

As shown in Figure 1, there are *marked* differences between photographic and medical images. Photographic images, especially those in ImageNet, depict a single object in the

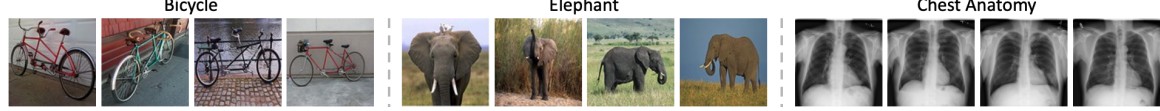

Figure 1: Photographic images typically have objects with discriminative parts (e.g., wheels of a bicycle) at the center, enabling instance discrimination methods to learn generalizable representations; medical images generated from a particular imaging protocol exhibit remarkable resemblance in anatomy (e.g., lungs), hindering these methods from capturing distinct features, resulting in poor transferability. To address this issue, this paper empowers various instance discrimination methods with the fine-grained, local context information embedded in medical images.

center of the image and also have discriminative visual features, e.g., wheels in a bicycle or tusks in an elephant. Hence, in the case of photographic images, a discriminative SSL approach that focuses solely on the most key discriminative features in the image (i.e., wheels in bicycle) could achieve high performance on the instance discrimination task (Gansbeke et al., 2021). By contrast, medical images (e.g. chest radiographs) display great similarities in anatomy with subtle differences in terms of organ shapes, boundaries, and texture (see examples in Figure 1). This naturally raises the following question: "*How well can instance discrimination methods extract generalizable features when applied to medical images?*"

We approach this question by pretraining recent state-of-the-art (SOTA) instance discrimination methods, with diverse learning objectives, on unlabeled chest X-ray images. We empirically found that instance discrimination methods may not learn a *distinct* set of features from medical images, which results in a negative impact on the generality of their features for various downstream tasks (see Table 4). This makes intuitive sense because these methods define their objectives based on a global representation of the images, overlooking important visual details in smaller local regions. Hence, such global representations may not be sufficient to distinguish medical images, which render similar global structures, from one another. Therefore, we suspect that instance discrimination methods rely on "superficial" features to distinguish individual medical images (e.g., X-rays in Figure 1), resulting in poor generalizability; we hypothesize that finer detailed information embedded in the local context of medical images can serve as a *philosopher's stone* for instance discrimination methods, assisting them in extracting more discriminative and diverse features from medical images. As a result, we ponder "*Can we enhance instance discrimination self-supervised learning by encapsulating context-aware representations?*"

Unsupervised generative tasks in different domains, including vision (Pathak et al., 2016), text (van den Oord et al., 2019), audio (Schneider et al., 2019), and medical (Haghighi et al., 2020, 2021), have shown great promise in exploiting spatial context as a powerful source of automatic supervisory signal for squeezing out rich representation. Thus, we propose Context-Aware instance Discrimination (CAiD), a simple yet powerful self-supervised framework that formulates an auxiliary context prediction task to equip instance discrimination learning with *context-aware representations*. To verify our hypothesis, we take three

representative recent SOTA self-supervised methods with varying discrimination objectives: MoCo-v2 (Chen et al., 2020b), Barlow Twins (Zbontar et al., 2021), and SimSiam (Chen and He, 2021), and couple them with a generative task in our end-to-end framework. Our extensive experiments reveal that CAiD (1) enriches representations learned from existing instance discrimination methods, yielding more informative and diverse visual representations; (2) outperforms two fully-supervised baselines— including (*de facto*) ImageNet and (competitive) in-domain pre-trained models, setting a new SOTA for self-supervised learning in medical imaging; (3) provides more discriminative features by adequately capturing finer contextual information from individual medial images, separating them effectively apart; and (4) enhances reusability of low/mid-level features when compared to standard instance discrimination methods, leading to higher transferability to different tasks.

To the best of our knowledge, this is the first work that quantitatively and systematically shows the limitation of instance discrimination methods in learning a distinct set of features from medical images and that offers a solution for alleviating the unearthed limitation. In summary, we make the following main contributions:

- A deep analysis that yields several new insights into the general limitation of existing instance-based objectives in capturing a set of distinct features from unlabeled medical images due to their fundamental anatomical similarity.

- A novel and scalable self-supervised learning framework that elevates existing instance discrimination methods for medical imaging.

- A new and comprehensive evaluation strategy that opens up novel perspectives for analyzing representation learning from various viewpoints, including (i) feature transferability, (ii) feature separation, and (iii) feature reuse, yielding new insights for developing more advanced SSL methods for medical imaging.

### 1.1. Related works

**Instance discrimination SSL** methods spark a renaissance in the SSL paradigm. These methods seek to learn invariant representations to image transformations and can be categorized into three groups based on their learning objectives: *contrastive learning* (He et al., 2020; Chen et al., 2020a), *information maximization* (Zbontar et al., 2021), and *distillation* (Chen and He, 2021; Grill et al., 2020). Despite their great success, these methods learn a global representation of images, limiting their generalization to the tasks that require finer-grained representations (Xie et al., 2021b,a), such as medical applications.
**Context prediction SSL** methods utilize image context as a rich source of information to formulate *discriminative* (Noroozi and Favaro, 2016) or *generative* (Pathak et al., 2016) pretext tasks; the former uses spatial context to generate pseudo labels for training a discriminator model, while the latter uses a generator model to restore the perturbed context.
**SSL in medical imaging** has made significant progress in recent years. While early efforts relied heavily on context prediction, recent years have witnessed a growing interest in instance discrimination approaches, particularly contrastive-based SSL methods (Azizi et al., 2021; Zhou et al., 2020; Kaku et al., 2021; Chaitanya et al., 2020). The recent transfer learning benchmark in medical imaging (Hosseinzadeh Taher et al., 2021) revealed the

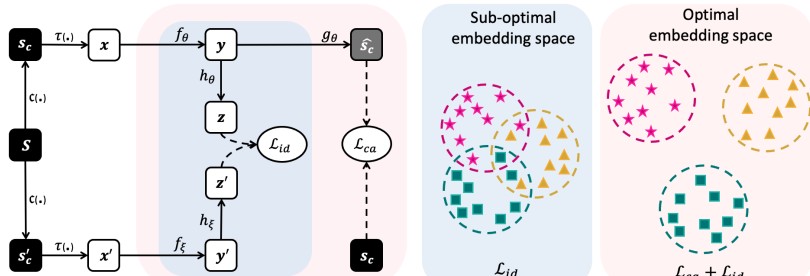

Figure 2: **An overview of the CAiD framework.** Towards learning an optimal embedding space with more discriminative features for medical images, we incorporate a context-aware representation learning with instance discrimination learning. The instance discrimination branch maximizes the (feature-level) similarity between the representations of augmented views $x$ and $x'$. The context learning branch maximizes the (pixel-level) similarity between original sample $s_c$ and restored $\hat{s}_c$.

potency of a variety of instance discrimination methods pre-trained on ImageNet for diverse medical tasks. Recently, TransVW (Haghighi et al., 2021) proved the efficacy of integrating discriminative and generative components into a single SSL framework for medical imaging. Motivated by the success of TransVW, PCRL (Zhou et al., 2021) proposed a reconstruction task to encode more information into the representations learned from the contrastive loss. Our contributions depart from this line of approach by showing two significant advances: (1) providing the first systematical analysis regarding the limitations of instance discrimination learning for medical imaging; (2) elucidating the importance of fine-grained contextual information in elevating a diverse set of instance-based objectives in medical image analysis.

## 2. CAiD Framework

Given the fundamental anatomical similarity of medical images (see Figure 1), the global representations captured by standard instance discrimination methods may be insufficient to distinguish medical images from each other. In fact, such coarse-grained representations may lead to a sub-optimal embedding space, which does not generalize well to different downstream tasks. Towards an optimal embedding space, our SSL approach exploits the diversity in the local context of images to empower instance discrimination learning with more discriminative features, distinguishing individual images more effectively. As shown in Figure 2, CAiD integrates two key components:

**(1) Instance Discrimination Learning** aims to maximizes the similarity of representations obtained from different augmented views of an image. Given a sample $S$, we first apply a random cropping operator $c(.)$ on $S$ to obtain two image crops $s_c$ and $s'_c$. The two crops are then augmented by applying an augmentation operator $\tau(.)$, resulting in two augmented views $x$ and $x'$. Next, $x$ and $x'$ are encoded by two encoder networks $f_\theta$ and $f_\xi$ into latent representations $y = f_\theta(x)$ and $y' = f_\xi(x')$. $y$ and $y'$ are further projected by the projector heads $h_\theta$ and $h_\xi$ to generate projections $z = h_\theta(y)$ and $z' = h_\xi(y')$. The discrimination

loss maximizes the similarity between $z$ and $z'$, and has a general form of $\mathcal{L}_{id} = sim(z, z')$, where $sim(.)$ is a similarity function that measures agreement between $z$ and $z'$. Generally, CAiD is applicable to any instance discrimination method. As such, while $f_\theta$ is a regular encoder, $f_\xi$ can be a momentum encoder (He et al., 2020) or share weights with $f_\theta$ (Zbontar et al., 2021); $sim(.)$ can be contrastive loss (He et al., 2020), cosine similarity (Chen and He, 2021), redundancy reduction loss (Zbontar et al., 2021), etc.

**(2) Context-aware Representation Learning** aims to assist instance discrimination learning by encoding finer and discriminative information from local context of images. To do so, given the image crop $s_c$ augmented by $\tau(.)$, the encoder network $f_\theta$ and decoder network $g_\theta$ are optimized to learn a mapping from the augmented crop to the original one, i.e., $f_\theta, g_\theta : (s_c, \tau) \mapsto s_c$. Through reconstructing the missing or corrupted image crops, the model is forced to learn context-aware representations, capturing the diversity of intensity, shape, boundary, and texture among images. The auxiliary context-aware learning loss maximizes the similarity between original crop and the reconstructed one, and has a general form of $\mathcal{L}_{ca} = sim(s_c, \hat{s}_c)$, where $\hat{s}_c = g_\theta(f_\theta(\tau(s_c)))$ presents the reconstructed crop. $sim(.)$ measures similarity between $s_c$ and $\hat{s}_c$ and can be $L_1$ or $L_2$ distance, etc.

**Integrated Objective.** Our approach integrates both learning schemes and jointly train them with an overall loss $\mathcal{L} = \lambda * \mathcal{L}_{ca} + \mathcal{L}_{id}$, where $\lambda$ is a constant weight for trading off the importance of each term of the loss. To solve this task, the model must encode local contextual information about the image while making the representation invariant to the augmentation applied to the image, leading to more discriminative and diverse features.

## 3. Experiments and Results

**Pre-training setup.** We apply CAiD to three recent SOTA SSL methods with different discrimination objectives: MoCo-v2, Barlow Twins, and SimSiam. For each method, we follow the original paper in the formulation of $\mathcal{L}_{id}$, projector head architecture, optimization setups, and hyper-parameters settings (details in Section A.1). We use U-Net with a standard ResNet-50 backbone as the $f_\theta$ and $g_\theta$, batch size 256, and $L2$ distance as the $\mathcal{L}_{ca}$. All models are pretrained from scratch on training set of ChestX-ray14 (Wang et al., 2017) dataset. $\lambda$ is set to 10. Images are resized to 224×224; augmentations include random horizontal flipping, color jittering, and Gaussian blurring. Moreover, we apply cutout (DeVries and Taylor, 2017) and shuffling (Chen et al., 2019) to enhance context learning.

**Transfer learning setup.** We evaluate the effectiveness of CAiD representations in transfer learning to a diverse set of four challenging medical imaging tasks, including classification on ChestX-ray14 and CheXpert (Irvin et al., 2019), and segmentation on SIIM-ACR (PNE, 2019) and NIH Montgomery (Jaeger et al., 2014) datasets (details in Section B). We transfer pre-trained (1) encoder to the classification tasks, and (2) encoder and decoder to segmentation tasks. We fine-tune all the parameters of downstream models (details in Section A.2).

### 3.1. Transfer Learning to Downstream Tasks

**A) CAiD enriches existing instance discrimination methods.**
**Experimental setup.** To assess the efficacy of our CAiD in enriching instance discrimination methods, we apply it to Barlow Twins, MoCo-v2, and SimSiam. All CAiD and original methods are pre-trained on the ChestX-ray14 dataset and fine-tuned on downstream tasks.

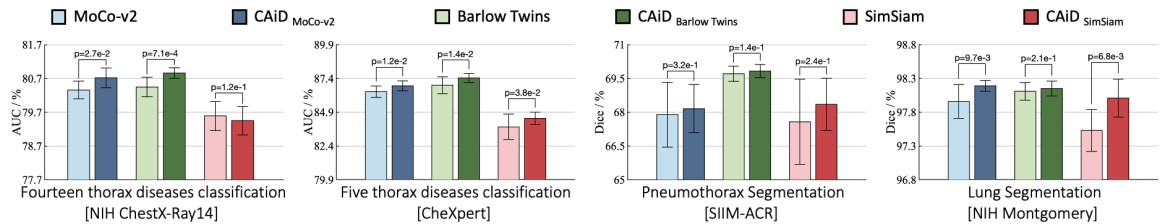

Figure 3: **Comparison with instance discrimination SSL methods:** CAiD empowers instance discrimination methods to capture more generalizable representations, yielding significant ($p < 0.05$) performance gains on four downstream tasks.

Table 1: **Comparison with fully-supervised transfer learning:** CAiD models outperform fully-supervised pre-trained models in 3 downstream tasks. The ‡ and † present the statistically significant ($p < 0.05$) and equivalent performances, respectively, compared to supervised ImageNet and ChestX-ray14 baselines.

| Initialization | Pre-training dataset | Classification [AUC (%)] | | Segmentation [Dice (%)] | |
|---|---|---|---|---|---|
| | | ChestX-ray14 | CheXpert | SIIM-ACR | Montgomery |
| Random | - | 80.31±0.10 | 86.62±0.15 | 67.54±0.60 | 97.55±0.36 |
| Supervised | ImageNet | **81.70±0.15** | 87.17±0.22 | 67.93±1.45 | 98.19±0.13 |
| Supervised | ChestX-ray14 | - | 87.40±0.26 | 68.92±0.98 | 98.16±0.05 |
| CAiD$_{\text{MoCo-v2}}$ | ChestX-ray14 | 80.72±0.29 | 86.86±0.37 † † | 68.16±1.07 † † | **98.19±0.08** † † |
| CAiD$_{\text{Barlow Twins}}$ | ChestX-ray14 | 80.86±0.16 | **87.44±0.33** ‡ † | **69.83±0.29** ‡ ‡ | 98.15±0.11 † |
| CAiD$_{\text{SimSiam}}$ | ChestX-ray14 | 79.45±0.42 | 84.45±0.46 | 68.35±1.16 † † | 98.01±0.28 † |

**Results.** As shown in Figure 3, CAiD improves the underlying instance discrimination methods across all tasks, yielding robust performance gains on both classification and segmentation tasks. Compared with original methods, CAiD$_{\text{MoCo-v2}}$ and CAiD$_{\text{Barlow Twins}}$ provide improved performance in all downstream tasks; CAiD$_{\text{SimSiam}}$ shows increased performance on three tasks and equivalent performance with SimSiam in ChestX-ray14.

**B) CAiD outperforms fully-supervised pre-trained models.**

**Experimental setup.** Following the recent transfer learning benchmark for medical image analysis (Hosseinzadeh Taher et al., 2021), we compare the transferability of representations learned by our CAiD models, which were pre-trained solely on unlabeled chest X-rays, with fully-supervised pre-trained models on ImageNet (the most common baseline) and ChestX-ray14 (the upper-bound in-domain baseline).

**Results.** As shown in Table 1, our CAiD models provide superior or on-par performance to both supervised baselines. CAiD$_{\text{Barlow Twins}}$ outperforms both supervised methods on CheXpert and SIIM-ACR; CAiD$_{\text{MoCo-v2}}$ outperforms ImageNet on SIIM-ACR and both baselines on Montgomery; CAiD$_{\text{SimSiam}}$ outperforms ImageNet on SIIM-ACR. These results demonstrate that our framework, with zero annotated data, is capable of providing more pronounced representation compared with supervised pre-training, confirming its potential for reducing medical imaging annotation costs.

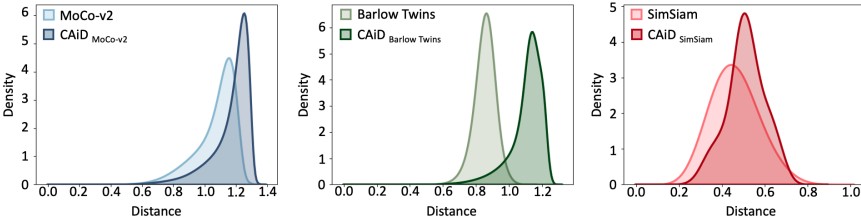

Figure 4: **Comparison of feature distance distributions.** CAiD enlarged feature distances compared with the original instance discrimination methods.

### 3.2. Feature Analysis

**A) CAiD provides more separable features.** Instance discrimination SSL methods aim to learn an optimal embedding space in which all instances are well-separated. Improved separation of images in an embedding space implies that the SSL method has learned more discriminative features, leading to better generalization to different tasks (Islam et al., 2021).
**Experimental setup.** Following (Chen et al., 2021b; Wang and Isola, 2020), we compute the distribution of distances between features learned by our CAiD approach, and compare it with the original instance discrimination counterpart. To do so, we first use the pre-trained models to extract features of the ChestX-ray14's test images. We extract features from the last layer of the ResNet-50 backbone and pass them to a global average pooling layer to obtain a feature vector for each images. Then, we normalize the features and compute all pairwise distances between features of individual images using the Euclidean distance metric. Finally, we visualize the distance distributions with Gaussian kernel density estimation (KDE). An SSL method that captures more diverse and discriminative representations, yields an embedding space with larger feature distances.
**Results.** As seen in Figure 4, CAiD provides substantially increased feature distances; the mean distance of CAiD$_{\text{MoCo-v2}}$, CAiD$_{\text{Barlow Twins}}$, and CAiD$_{\text{SimSiam}}$ increased by 9%, 30%, and 11% in comparison with the original methods, respectively. We conclude CAiD delivers more discriminative features by adequately capturing finer contextual information from individual images, separating them apart effectively.
**B) CAiD provides more reusable low/mid-level features.** It is widely understood that convolutional neural networks build feature hierarchies– deep network lower layers control general low/mid-level features whereas higher layers contain more task-specific features (Neyshabur et al., 2020; Haghighi et al., 2021). The benefits of SSL are generally believed to stem from the reuse of pre-trained low/mid-level features in downstream tasks (Asano et al., 2020; Zhao et al., 2021; Islam et al., 2021). Higher feature reuse implies that a self-supervised model learns more useful features, leading to higher performance in downstream tasks, especially those with limited labeled data (Kaku et al., 2021).
**Experimental setup.** Following (Neyshabur et al., 2020), we use the Centered Kernel Alignment (CKA) (Kornblith et al., 2019) metric to investigate how our SSL approach can improve the feature reuse compared with the original instance discrimination methods. The CKA score shows the similarity of the features before and after fine-tuning on downstream

Table 2: **Comparison of feature reuse between CAiD and original instance discrimination methods.** Each row presents the CKA score for different intermediate layers before and after fine-tuning models in two downstream tasks.

| Method | Pre-training dataset | ChestX-ray14 | | | | | Montgomery | | | | |
|---|---|---|---|---|---|---|---|---|---|---|---|
| | | Layer 1 | Layer 2 | Layer 3 | Layer 4 | Layer 5 | Layer 1 | Layer 2 | Layer 3 | Layer 4 | Layer 5 |
| MoCo-v2 | ChestX-ray14 | 0.995 | 0.622 | 0.625 | 0.500 | 0.409 | 0.995 | 0.618 | 0.695 | 0.546 | 0.339 |
| $CAiD_{MoCo-v2}$ | ChestX-ray14 | **0.998** | **0.874** | **0.739** | 0.571 | 0.354 | **0.997** | **0.864** | **0.754** | 0.455 | 0.272 |
| Barlow Twins | ChestX-ray14 | 0.948 | 0.709 | 0.613 | 0.787 | 0.482 | 0.916 | 0.627 | 0.779 | 0.745 | 0.297 |
| $CAiD_{Barlow\ Twins}$ | ChestX-ray14 | **0.992** | **0.896** | **0.747** | 0.808 | 0.498 | **0.985** | **0.910** | **0.876** | 0.682 | 0.403 |
| SimSiam | ChestX-ray14 | 0.977 | 0.629 | 0.677 | 0.683 | 0.480 | 0.978 | 0.527 | 0.577 | 0.532 | 0.402 |
| $CAiD_{SimSiam}$ | ChestX-ray14 | **0.993** | **0.901** | **0.724** | 0.557 | 0.462 | **0.995** | **0.917** | **0.778** | 0.509 | 0.380 |

tasks; if an SSL pre-trained model provides features that are similar to the fine-tuned model, it indicates that the SSL approach has learned more useful features. Following (Kaku et al., 2021), we evaluate feature reuse in small labeled data regimes on classification (10% labeled data of the ChestX-ray14) and segmentation (Montgomery) tasks. We extract features from conv1 and the end of four residual blocks of ResNet-50 backbone, denoted as layers 1 to 5, and then pass them to a global average pooling layer to compute feature similarity. We fine-tune each method multiple times and report average CKA score on each task.

**Results.** Each row of Table 2 presents the per-layer feature similarity between a pre-trained model and the corresponding fine-tuned model. We observe CAiD models consistently provide highly reusable low/mid-level features (layers 1 to 3) compared with the original discriminative methods: $CAiD_{MoCo-v2}$, $CAiD_{Barlow\ Twins}$, and $CAiD_{SimSiam}$ lead to an average gain of 12%, 12%, and 11% across the first three layers in the classification task, and 10%, 15%, and 20% in segmentation task. These results indicate that encoding context-aware representations lead to more reusable features that generalize better to downstream tasks with low-data regimes. Furthermore, we observe that the initial layers provide more reusable features compared to the higher layers (i.e. layers 4 and 5). This observation, consistent with (Asano et al., 2020; Zhao et al., 2021) and in accordance with our transfer learning results, demonstrates that low/mid level features are truly important for transfer learning.

## 4. Conclusion

We have investigated the applicability of instance discrimination SSL methods in medical imaging, revealing that the high global anatomic similarity of medical images hinders these methods from learning a distinct set of features essential for medical tasks. To alleviate this problem, we have developed CAiD to enhance instance discrimination learning with more discriminative features by leveraging diversity in the local context of images via a generative task. Our feature analysis reveals that learning a holistic encoding over the entire medical image, using a generative task, encourages the instance discrimination approach to effectively distinguish medical images from one another, resulting in a more discriminative feature space. Our extensive experiments also show that, when compared to standard instance discrimination methods, our training schema can effectively improve the reusability of low/mid-level features, resulting in greater transferability to different medical tasks. In the future, we plan to optimize $\mathcal{L}_{ca}$ to enhance our context learning approach.

## Acknowledgments

This research has been supported in part by ASU and Mayo Clinic through a Seed Grant and an Innovation Grant and in part by the NIH under Award Number R01HL128785. The content is solely the responsibility of the authors and does not necessarily represent the official views of the NIH. This work has utilized the GPUs provided in part by the ASU Research Computing and in part by the Extreme Science and Engineering Discovery Environment (XSEDE) funded by the National Science Foundation (NSF) under grant number ACI-1548562. The content of this paper is covered by patents pending.

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

## Appendix A. Implementation

### A.1. Pre-training Settings

We apply CAiD to three popular instance discrimination methods, including MoCo-v2, Barlow Twins, and SimSiam, which serve as the basis for our empirical evaluation. These methods share the approach of encoding two augmented image views using two backbone encoders and projection heads, and maximizing the agreement between their representations. For completeness, we outline each method below and provide additional pre-training details complementing data presented in Section 3.

**MoCo-v2** (Chen et al., 2020b): MoCo-v2 is a popular representative of contrastive learning methods. It minimizes the positive pair distances, while maximizing the negative pair distances. Positive pairs consists of different augmented views of the same image, whereas negative pairs are other images. To benefit from sufficient negative pairs, a queue $K = \{k_1, k_2, ...k_N\}$ is utilized to store the representations of negative samples. Moreover, MoCo leverages a momentum encoder to ensure the consistency of negative samples as they evolve during training. When adopting MoCo-v2 in CAiD, the encoder $f_\theta$ and projection head $h_\theta$ are updated by back-propagation, while $f_\xi$ and $h_\xi$ are updated by using an exponential moving average (EMA) of the parameters in $f_\theta$ and $h_\theta$, respectively. The loss function is contrastive loss (van den Oord et al., 2019), which for a pair of positive samples $x$ and $x'$ is defined as follows:

$$\mathcal{L}_{id} = -log\frac{exp(z \cdot z'/\tau)}{\sum\limits_{n=0}^{N} exp(z \cdot k_n/\tau)} \tag{1}$$

where $z = h_\theta(f_\theta(x))$ and $z' = h_\xi(f_\xi(x'))$, $\tau$ is a temperature hyper-parameter, and $N$ is the queue size. Following (Chen et al., 2020b), we use a standard ResNet-50 as $f_\theta$ and a two-layer MLP head (hidden layer 2048-d, with ReLU) as $h_\theta$ . Additionally, $f_\theta$, $h_\theta$, and $g_\theta$ are optimized using SGD with an initial learning rate of 0.03, weight decay 0.0001, and the SGD momentum 0.9.

**Barlow Twins** (Zbontar et al., 2021): Barlow Twins is a popular and effective representative of redundancy reduction instance discrimination learning methods. Barlow Twins makes the cross-correlation matrix computed from two Siamese branches close to the identity matrix. By equating the diagonal elements of the cross-correlation matrix to 1, the representation will be invariant to the distortions applied to the samples. By equating the off-diagonal elements of the cross-correlation matrix to 0, the different vector components of the representation will be decorrelated, so that the output units contain non-redundant information about the sample. The discrimination loss is defined as follows:

$$\mathcal{L}_{id} = \sum_i (1 - \mathcal{C}_{ii})^2 + \lambda \sum_i \sum_{i \neq j} \mathcal{C}_{ij}^2 \tag{2}$$

where $\mathcal{C}$ is the cross-correlation matrix computed between the outputs of the $h_\theta$ and $h_\xi$ networks along the batch dimension. $\lambda$ is a coefficient that determines the importance of the invariance term and redundancy reduction term in the loss. Following (Zbontar et al., 2021), $f_\theta$ is a standard ResNet-50 and $h_\theta$ is a three-layer MLP head. Moreover, when adopting

Barlow Twins in CAiD, $f_\theta/h_\theta$ share weights with $f_\xi/h_\xi$. $f_\theta$, $h_\theta$, and $g_\theta$ are optimized using LARS optimizer with the learning rate schedule described in (Zbontar et al., 2021).

**SimSiam** (Chen and He, 2021): SimSiam is a recent representative of asymmetric instance discrimination methods. SimSiam directly maximizes the similarity of two views from an image using a simple siamese network followed by a predictor head, omitting the negative pairs in contrastive learning. A stop-gradient operation is leveraged to prevent collapsing solutions. Specifically, the model parameters are only updated using one distorted version of the input, while the representations from another distorted version are used as a fixed target. The model is trained to maximize the agreement between the representations of positive samples using negative cosine similarity, defined as follows:

$$\mathcal{D}(z, y') = -\frac{z}{\|z\|_2} \cdot \frac{y'}{\|y'\|_2} \tag{3}$$

where $z = h_\theta(f_\theta(x))$ and $y' = f_\xi(x')$. The discrimination branch is trained using a symmetrized loss as follows:

$$\mathcal{L}_{id} = \frac{1}{2}\mathcal{D}(z, stopgrad(y')) + \frac{1}{2}\mathcal{D}(z', stopgrad(y)) \tag{4}$$

where stopgrad means that $y'$ is treated as a constant in this term. Following (Chen and He, 2021), $f_\theta$ is a standard ResNet-50 and $h_\theta$ is a three-layer projection MLP head (hidden layer 2048-d), followed by a two-layer predictor MLP head. Moreover, when adopting SimSiam in CAiD, $f_\theta$, $h_\theta$, and $g_\theta$ are optimized using SGD with a linear scaling learning rate (lr×BatchSize/256). The initial learning rate is 0.05, weight decay is 0.0001, and the SGD momentum is 0.9.

**Full training process:** Following (Chaitanya et al., 2020; Chen et al., 2021a), we start with training the instance discrimination task to warm up the model; the encoder $f_\theta$ along with projector $h_\theta$ are optimized using $\mathcal{L}_{id}$ following the learning schedule of the original methods (Chen et al., 2020b; Chen and He, 2021; Zbontar et al., 2021), enabling the model with an initial discrimination ability. The context representation learning loss is then added and the whole network is trained jointly using $\lambda * \mathcal{L}_{ca} + \mathcal{L}_{id}$; the optimization of the framework by incorporation of $\mathcal{L}_{ca}$ takes up to 400 epochs. Following (Zhou et al., 2021; Haghighi et al., 2021; Kaku et al., 2021), the checkpoints with the lowest validation loss are used for fine-tuning.

### A.2. Fine-tuning Settings

We use AUC (area under the ROC curve) and Dice coefficient for measuring the accuracy of classification and segmentation tasks, respectively. We endeavor to optimize each downstream task with the best performing hyper-parameters. In all downstream tasks, we employ the early-stop mechanism using 10% of the training data as the validation set to avoid overfitting. We run each method ten times on each downstream task and report the average, standard deviation, and further present statistical analysis based on an independent two-sample $t$-test. All pre-training methods benefit from the same network architecture, data preprocessing and augmentation, and optimization setup in all downstream tasks, described in the following.

**Network architecture.** In the classification downstream tasks, the standard ResNet-50 encoder followed by a task-specific classification head is used. In the segmentation downstream tasks, a U-Net[1] network with a ResNet-50 encoder is utilized.

**Preprocessing and data augmentation:** Following (Hosseinzadeh Taher et al., 2021), in all downstream tasks, we resize the images to 224×224. For thoracic diseases classification tasks on ChestX-ray14 and CheXpert, we apply standard data augmentation techniques, including random crop and resize, horizontal flip, and rotation. For segmentation tasks on SIIM-ACR and Montgomery, we apply random brightness contrast, random gamma, optical distortion, elastic transformation, and grid distortion.

**Optimization:** We endeavour to optimize each downstream task with the best performing hyper-parameters. For all downstream tasks, we use Adam optimizer with $\beta_1 = 0.9$, $\beta_2 = 0.999$. We use early-stop mechanism using the 10% of the training data as the validation set to avoid over-fitting. For classification tasks on ChestX-ray14 and CheXpert datasets, we use a learning rate $2e-4$ and *ReduceLROnPlateau* as the learning rate decay scheduler. For segmentation tasks on SIIM-ACR and Montgomery, we use a learning rate $1e-3$ and *cosine* learning rate decay scheduler.

## Appendix B. Datasets and Downstream Tasks

We evaluate the effectiveness of our CAiD representations in transfer learning to a diverse set of four popular and challenging medical imaging tasks on chest X-ray images. These tasks cover both the downstream tasks on the same dataset as pre-training and downstream tasks with a variety of domain shifts in terms of data distribution and disease/object of interest. In this section, we provide the details of each dataset and the underlying task, as well as the evaluation metric for each task.

**ChestX-ray14:** ChestX-ray14 is a hospital-scale publicly-available dataset, including 112K chest X-ray images taken from 30K unique patients. The ground truth consists of 14 thoracic disease labels associated with each image. We use the official patient-wise split released with the dataset, including 86K training images and 25K testing images. Training images without label are used for pre-training of our models, whereas labels are used only in downstream tasks for evaluating transfer learning. Downstream task on this dataset is a multi-label classification task; the models are trained to predict 14 thoracic pathologies. We report the mean AUC score over 14 pathologies to evaluate the classification accuracy.

**CheXpert:** CheXpert is a hospital-scale publicly available dataset, including 224K chest X-ray images taken from 65K unique patients. The ground truth for the training set consists of 14 thoracic disease labels associated with each image, which were obtained automatically from radiology reports. The testing set's ground truths were obtained manually from board-certified radiologists, including 5 selected thoracic pathologies— cardiomegaly, edema, consolidation, atelectasis, and pleural effusion. We use the official data split released with the dataset, including 224K training and 234 test images. Downstream task on this dataset is a multi-label classification task; the models are trained to predict five pathologies in a multi-label classification setting. We report the mean AUC score over 5 pathologies to evaluate the classification accuracy.

---

1. Segmentation Models: https://github.com/qubvel/segmentation_models.pytorch

Table 3: **Transfer learning under different downstream label fractions:** CAiD models provide more generalizable representations for downstream tasks with limited annotated data compared with the original instance discrimination methods.

| Method | Pre-training dataset | ChestX-ray14 [AUC (%)] | |
| --- | --- | --- | --- |
| | | Label fraction | |
| | | 10% | 25% |
| MoCo-v2 | ChestX-ray14 | 67.17±0.99 | 74.89±1.03 |
| CAiD$_{\text{MoCo-v2}}$ | ChestX-ray14 | **70.00±1.37** | **75.25±0.48** |
| Barlow Twins | ChestX-ray14 | 73.14±1.81 | 76.23±0.60 |
| CAiD$_{\text{BarlowTwins}}$ | ChestX-ray14 | **73.92±1.26** | **77.23±1.01** |
| SimSiam | ChestX-ray14 | 67.28±1.09 | 73.05±1.23 |
| CAiD$_{\text{SimSiam}}$ | ChestX-ray14 | **67.34±0.98** | **73.75±1.10** |

**SIIM-ACR:** The dataset is provided by the Society for Imaging Informatics in Medicine (SIIM) and American College of Radiology. It consists of 10K chest X-ray images and pixel-wise segmentation mask for pneumothorax disease. We randomly divided the dataset into a training (80%) and testing (20%) set. Downstream task on this dataset is a pixel-level segmentation task; models are trained to segment pneumothorax within chest X-ray images (if present). We report the mean Dice coefficient score to evaluate the segmentation accuracy.

**NIH Montgomery:** This publicly available dataset is provided by the Montgomery County's tuberculosis screening program. The dataset provides 138 chest X-ray images, including 80 normal cases and 58 cases with tuberculosis (TB) indications in this dataset. Moreover, ground truth segmentation masks for left and right lungs are provided. We randomly divided the dataset into a training set (80%) and a test set (20%). Downstream task on this dataset is a pixel-level segmentation task; models are trained to segment left and right lungs in chest X-ray images. We report the mean Dice coefficient score to evaluate the segmentation accuracy.

## Appendix C. Transfer Learning to Small Data-regimes

**Experimental setup.** We further investigate the robustness of representations learned with our CAiD in the small data regimes. To do so, we follow the common protocol (Azizi et al., 2021; Haghighi et al., 2021) and randomly select 10% and 25% of labeled training data from ChestX-ray14 dataset, and fine-tune the self-supervised pre-trained models on these training-data fractions using the previously explained fine-tuning protocol. We run each method ten times and report the average performance.

**Results.** Table B summarizes the results. As seen, CAiD pre-trained models achieve superior performance in all data subsets compared with the original instance discrimination methods. Specifically, Compared to the original methods, CAiD$_{\text{MoCo-v2}}$ showed increased performance by 2.83% and 0.3% when using 10% and 25% of labeled data, respectively. Similarly, CAiD$_{\text{BarlowTwins}}$ showed increased performance by 0.78% and 1%. Finally, CAiD$_{\text{SimSiam}}$ showed increased performance by 0.06% and 0.7% when fine-tuning on 10% and 25% of labeled data, respectively. Our results demonstrate that our framework

Table 4: Comparison of instance discrimination methods, pretrained on chest X-ray images, with training from scratch. In each downstream task, we run each method ten times and conduct the statistical analysis between random initialization and each self-supervised method. Instance discrimination SSL methods perform very poorly, as they offer performance equally or even worse than training from scratch on some tasks. We attribute this inferior performance to the high global similarity of medical images in terms of anatomy, which prevents instance discrimination methods from learning a distinct set of features required for medical tasks.

| Initialization | ChestX-ray14 | | CheXpert | | SIIM-ACR | | Montgomery | |
|---|---|---|---|---|---|---|---|---|
| | Performance | $p-$value | Performance | $p-$value | Performance | $p-$value | Performance | $p-$value |
| Random | $80.31\pm0.10$ | - | $86.62\pm0.15$ | - | $67.54\pm0.60$ | - | $97.55\pm0.36$ | - |
| MoCo-v2$_{\text{X-rays}}$ | $80.36\pm0.26$ | 3.46e-1 | $86.42\pm0.42$ | 1.20e-1 | $67.89\pm1.44$ | 4.27e-1 | $97.96\pm0.25$ | 4.69e-3 |
| Barlow Twins$_{\text{X-rays}}$ | $80.45\pm0.29$ | 4.71e-1 | $86.90\pm0.62$ | 3.33e-1 | $69.71\pm0.34$ | 2.20e-4 | $98.11\pm0.13$ | 3.81e-4 |
| SimSiam$_{\text{X-rays}}$ | $79.59\pm0.43$ | 2.68e-3 | $83.82\pm0.94$ | 8.65e-4 | $67.58\pm1.16$ | 1.00e-1 | $97.53\pm0.31$ | 4.48e-1 |

provides more generalizable representations for downstream tasks with limited annotated data, helping reduce the annotation cost.

## Appendix D. A Study of Instance Discrimination Methods

Our study is based on a preliminary analysis on instance discrimination methods. We pretrained recent SOTA methods with diverse learning objectives on unlabeled chest X-ray images. We then compare their transfer learning performance with training from scratch (random initialization). The results of this study is presented in Table 4. As seen, instance discrimination SSL methods present mixed gains in different tasks. In particular, in ChestX-ray14 and CheXpert datasets, all methods present equivalent or worse performance than training from scratch. On the other hand, in SIIM-ACR, Barlow Twins provides significant gains compared with training from scratch, while the other methods present equivalent performance with baseline. Finally, in Montgomery, Barlow Twins and MoCo-v2 provide significant gains compared with baseline, while SimSiam has comparable performance. Given these results, we argue that directly employing instance discrimination methods is not enough for learning sufficiently detailed information from medical images. This is because these methods define their objectives based on a global representation of the images, overlooking important visual details in smaller local regions. However, such global representations may not be sufficient to distinguish medical images, which render similar global structures, from one another, hampering instance discrimination methods in capturing a set of distinct features. Our CAiD addresses this limitation by leveraging fine-grained contextual information captured from the local regions within images. As demonstrated in our concurrent work (Haghighi et al., 2022), a more optimal formulation of $\mathcal{L}_{ca}$ provides stronger contextual representations, which assists the discriminative component more effectively. Therefore, in the future, examining various choices of context prediction tasks may lead to further improvements in CAiD models.

