# OpenReview forum: "CAiD: Context-Aware Instance Discrimination for Self-supervised Learning in Medical Imaging"
_MIDL.io/2022/Conference — MIDL 2022_

### Official Review · Reviewer_UoRX · 2022-01-23

**Confidence:** 5
**Preliminary Rating:** 2
**Recommendation:** Oral, Poster

**Summary:**

This submission deals with self-supervised learning and hypothesis that instance discrimination works only well in natural images and medical scans require context-aware SSL steps in addition. The authors claim to be the first to "alleviate" the limitation of limited global context in self-supervised learning for medical images. The experimental results demonstrate significant improvements when combining context and instance discrimination w.r.t. three commonly used instance discrimination methods (MoCo V2, Barlow Twins, SimSiam) on a number of downstream tasks.

**Strengths:**

- The outset of incorporating both local and global clues for self-supervision is well motivated.
- The experimental results demonstrates that combining the CAiD approach with the baselines MoCo V2, Barlow Twins, SimSiam helps increasing downstream accuracies.
- They even outperform supervised pre-training (although the task similarity seems problematic).
- The feature distribution analysis provides some interesting insights.

**Weaknesses:**

I definitely disliked the way the authors increased their manuscript length by moving important sections, first and foremost Related Work to the appendix. In my opinion a paper should be concise and self-comprehensive without additional 6 full pages of appendix. In the following my main concerns and limitations are listed (references below)

- The authors miss quite a bit of related work that e.g. employed and extended the concept of Doersch's context prediction for SSL to medical 3D scans [1]
They formulated a very similar assessment of the requirements of context in medical SSL: "here exists a conflicting relation between an increasing size of the patch for the CNN (equivalently its receptive field), which is necessary to capture enough spatial information for an expressive feature learning and a suitable difficulty of the auxiliary task: i.e. the learning task can become too easy, when the receptive field grows, because neighboring subvolumes are likely to contain easily identifiable structures"
- The authors also only mention all other related work on context pre-text tasks in passing (appendix) and do not perform any comparisons to Jigsaw, Rubik's cube etc. The excellent comprehensive approach of Zhou's Models Genesis is also missing.
- I found the interpretation of the superiority of the proposed SSL method compared to supervised ImageNet models a bit too wide-fetching. It may be true that "the most common transfer learning in medical imaging" but that does not make it a sensible choice as it is still severely affected by the problematic local-to-global relations in medical images that are different to natural images (as stated by the authors themselves)
- In fact, judging from Table 1 the supervised pre-training yields little gain compared to random initialisation and the proposed method itself only slightly improves the overall scores (significance was shown). In general the differences of employing more optimised architectures (just any current state-of-the-art approach for the given task) could easily outweigh those subtle improvements.
- I think an important limitation is that the authors considered a full fine-tuning of all layers and only give some limited details on the learning rate handling for different networks depth in the appendix. Other works on SSL have considered frozen feature extractors with only a linear layer that was continued with updates after transferring to the downstream task.
[1] How to learn from unlabeled volume data: Self-supervised 3d context feature learning
M Blendowski MICCAI 2019

**Deanonymize Review:**

yes

**Final Rating After The Rebuttal:**

4: Weak Accept

**Justification Of The Final Rating:**

The authors have very thoroughly addressed my queries and I am much more confident that this paper is now in great shape for publication. They nicely pointed out that ChestX-ray14 supervised pre-training and not just ImageNet is outperformed and they included the suggested additional comparisons to related work. The authors have convinced me that they have great expertise in the topic and will provide an insightful presentation/discussion of their method at MIDL 2022. The restructuring makes the paper more comprehensive in its first 8 pages. I believe they also answered the other reviewers' comments satisfactorily, although I couldn't check in detail because the programme chairs only gave reviewers a Saturday to read rebuttals and check the revised versions (this is of course not the fault of the authors).

**Paper Type:**

methodological development

**Questions To Address In The Rebuttal:**

I would like to know whether the paper can be restructured into being comprehensible within the 8-page limit and without going forth and back between important text in the appendix and somewhat superfluous content in the main paper.
I think it would be important to compare the CAiD approach to other context SSL methods. It would also be essential to start from state-of-the-art baselines. Otherwise the practical impact of subtle pre-training improvements is very limited.

**Special Issue:**

no

---

### Official Review · Reviewer_6Quq · 2022-01-24

**Confidence:** 4
**Preliminary Rating:** 5
**Recommendation:** Best Paper Award, Oral

**Summary:**

* A representation learning method is proposed.
* The main idea is to augment transformation-invariance (popular in the computer vision literature) with preservation of fine-grained information required for image reconstruction.
* The proposed pre-training consistently improves transfer learning performance across
    * 2 classification tasks
    * 2 segmentation tasks
    * 3 methods to promote transformation-invariance
* Analysis of the extracted features shows favourable properties in terms of improved pair-wise segregation and improved feature reuse for downstream tasks.
* I believe this paper would make for a stellar addition to the growing representation learning literature within the medical image analysis community, and could potentially attract a lot of discussion in the conference.

**Strengths:**

* The paper proposes a simple, well-motivated idea.
* The validation is extensive, and shows small but consistent improvements across the board.
* The analysis (feature segregation, reuse) is well done, and further showcases favourable properties of the method.
* The writing is extremely clear.

**Weaknesses:**

* The paper has no weaknesses, in my opinion. The conference paper is suitable for acceptance in its current state.

* If a journal extension is planned, the following comparisons should be included:
    * A comparison with existing work in the medical image analysis representation learning literature.
    * Image reconstruction is one task to encourage preservation of fine-grained information. A number of other self-supervised tasks could be used in its place. Indeed, authors do acknowledge this fact in the Appendix A. It would probably make sense to compare usage of different losses for $L_{CA}$ (e.g. denoising, inpainting).

**Deanonymize Review:**

no

**Final Rating After The Rebuttal:**

5: Strong Accept

**Justification Of The Final Rating:**

I really liked the proposed feature separation and reuse-based evaluation metrics and analysis.

In the rebuttal, the authors have added experimental comparisons with more SSL methods. Overall, in my opinion, the experiments sufficiently show the small, but consistent, improvements of the proposed method, when it is used in conjunction with any of several other methods.

**Paper Type:**

both

**Questions To Address In The Rebuttal:**

* Can the authors comment on the applicability of the proposed method to computer vision datasets? Potentially, retaining fine-grained information should be helpful there too. Can the authors envision a scenario where retaining such information could lead to worse performance in downstream tasks?

**Special Issue:**

yes

---

### Official Review · Reviewer_tsbr · 2022-01-25

**Confidence:** 4
**Preliminary Rating:** 2
**Recommendation:** Poster

**Summary:**

The paper presents a self-supervised learning framework called Context-Aware instance Discrimination (CAiD), which combines an instance discrimination technique such as MoCo, Barlow Twins or SimSiam with an image reconstruction loss that aims to preserve local context information in the representation. The proposed framework is used as pre-training method on two classification and two segmentation tasks from different datasets. Results show that it generally gives a higher improvement on downstream tasks, compared to the same instance discrimination technique without the reconstruction loss.

**Strengths:**

* The proposed approach of adding a generative loss based on reconstruction is simple but effective, as shown in the results.

* Experiments on the transferability and reusability of learned features are interesting and demonstrate the benefit of the proposed method.

* The paper is well written.


**Weaknesses:**

* Although the proposed method is well motivated, its novelty is somewhat limited. As I understand, the main technical contribution of this work is adding an "auto-encoder style" reconstruction loss to an existing instance discrimination SSL approach. However, this is a well-known technique for generative SSL (Liu et al., 2021). Moreover, the idea of combining instance discrimination and reconstruction-based SSL has already been explored in recent work like (Zhou et al., 2021).

* While the analysis of feature transferability and reusability is interesting, the experiments do not really demonstrate that the proposed method better captures the local context of images in the representation.

* Experiments only compare the method against instance discrimination baselines without the reconstruction loss, and does not include recent SSL approaches based on contrastive learning (Chaitanya et al., 2020) or prediction tasks like solving a jigsaw puzzle (Taleb et al., 2021) or inpainting (Tang et al., 2021).

* Results are not entirely convincing. The improvements of the added reconstruction loss on downstream tasks are often not statistically significant, which does not support the claim that instance discrimination techniques have important limitations.

Liu, X., Zhang, F., Hou, Z., Mian, L., Wang, Z., Zhang, J. and Tang, J., 2021. Self-supervised learning: Generative or contrastive. IEEE Transactions on Knowledge and Data Engineering.

Zhou, H.Y., Lu, C., Yang, S., Han, X. and Yu, Y., 2021. Preservational Learning Improves Self-supervised Medical Image Models by Reconstructing Diverse Contexts. In Proceedings of the IEEE/CVF International Conference on Computer Vision (pp. 3499-3509).

Chaitanya, K., Erdil, E., Karani, N. and Konukoglu, E., 2020. Contrastive learning of global and local features for medical image segmentation with limited annotations. arXiv preprint arXiv:2006.10511.

Taleb, A., Lippert, C., Klein, T. and Nabi, M., 2021, June. Multimodal self-supervised learning for medical image analysis. In International Conference on Information Processing in Medical Imaging (pp. 661-673). Springer, Cham.

Tang, Y., Yang, D., Li, W., Roth, H., Landman, B., Xu, D., Nath, V. and Hatamizadeh, A., 2021. Self-supervised pre-training of swin transformers for 3d medical image analysis. arXiv preprint arXiv:2111.14791.


**Deanonymize Review:**

no

**Detailed Comments:**

Other comments:

* While I understand that space is limited, I found it odd to have the Related Works section in the Appendix. Having this information in the introduction would help clarify the novel contributions of the paper.

* The proposed context-aware loss compares the original and reconstructed images. This is a good idea, but I wonder if comparing the features maps in the decoder would be more robust to differences caused by translation or shifts in intensity. This alternative technique was found to be efficient in (Chaitanya et al., 2021) and other recent works.

* The baselines in Table 1 and Table 2 are rather weak (e.g., pre-training with ImageNet). I recommend the authors to compare their method against a broader range of SSL techniques proposed for medical imaging.

* Section 3.2: " our framework, with zero annotation cost, is capable of providing more pronounced representation compared to supervised pre-training, showing its potential for reducing the annotation cost in medical imaging." This seems a little exaggerated: models pretrained on datasets like ImageNet are available at no additional cost.

* Figure 4: are the feature vectors normalized (e.g., unit norm)? Otherwise, their distance is not very meaningful.

* It would be useful to evaluate the performance obtained for different values of hyper-parameter lambda, and to visualize the reconstruction for these values.


**Final Rating After The Rebuttal:**

3: Borderline

**Justification Of The Final Rating:**

I appreciate the authors' detailed response to comments. However, my main concerns (regarding the limited technical contribution of the work and the fact that the advantages of the context-aware representation learning are not well demonstrated in experiments) were not fully addressed. Nevertheless, I have upgraded my score to Borderline.

**Paper Type:**

methodological development

**Questions To Address In The Rebuttal:**

I would like the authors to clarify and demonstrate the technical contributions of their work. Since the novelty revolves around the context-aware loss, it is necessary to better show the advantage of this loss and compare it against alternative techniques such as the local contrastive approach in (Chaitanya et al., 2021). Other comments are also important and need to be addressed.

**Special Issue:**

no

---

### Meta-Review · Area_Chair_5sce · 2022-02-18

**Recommendation:** Accept (Poster)
**Confidence:** 4

**Metareview:**

This paper proposes a self-supervised learning framework addressing issues specific to medical images, which are often globally similar but exhibit distinctive fine-grained differences. While reviewers' opinions were initially split about this paper, the authors responded very well to the reviewer concerns and substantially improved the paper, satisfying most of the reviewers' concerns and improving the overall rating to a solid recommendation for acceptance. The paper is well written, the method is well-motivated and thoroughly evaluated.

I agree with the reviewers and suggest to accept the paper for publication at MIDL.

---

### Decision · Program_Chairs · 2022-02-28

Accept